# Regulatory Evolution of Neonicotinoid Insecticides as Plant Protection Active Substances in Europe

Patrice A. Marchand 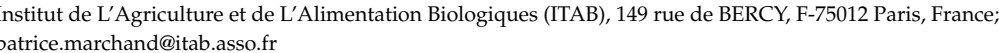

Institut de L'Agriculture et de L'Alimentation Biologiques (ITAB), 149 rue de BERCY, F-75012 Paris, France; patrice.marchand@itab.asso.fr

**Abstract:** Neonicotinoid (NN) insecticides derived from natural insecticide nicotine are EU chemical crop protection systemic active substances that are controversial regarding their toxicity and ecotoxicity, especially versus pollinators and birds. Clearly, the last European evaluation by the European Food Safety Authority exhibited a danger to wild and managed bees. Concomitantly, the decline in birds was partially attributed to this class of substances, which constitutes a family in itself, both in Europe and the USA. At the regulatory level, following the initial approval waves in 2011 and 2019, and mainly taking into account these ecotoxicological considerations, the commission banned the use of three NN insecticides in 2013 (imidacloprid, clothianidin, and thiamethoxam), and later, only one NN (acetamiprid) was renewed. Four NN approvals were removed by the end of approval or non-renewal in 2019 and 2020, and two are currently concerns for renewals in 2025, even if extensions of the approval durations of these NNs are to be expected due to the current slowness of the renewal procedure. Therefore, from the total number (17) of NN insecticide molecules known all over the world, up to seven NN were approved by the EU plant protection Regulation EC 1107/2009 between 2011 and now. All of these active substances are listed in Parts A and B of Regulation EU 540/2011 managing active substances. The regulatory evolution of these agrochemicals is analysed in this work, from the corresponding global modifications in terms of the number of active substances, employment, functions, uses covered, protected crops, and maximum residue limits. We also analysed their ability to persist as an agrochemical family and the potential of the inclusion of new NN members together with their current restrictions during their active substance life in Europe.

**Keywords:** plant protection products (PPP); neonicotinoid; ecotoxicology; risk assessment; agrochemicals

## 1. Introduction

Following our constant plant protection active substances (a.s.) survey, and especially agrochemicals [1–4], we recently focused on the evolution of chemical families [5]. Neonicotinoids (NNs) are active substances used as plant protection products (PPPs) with common insecticide functions [6,7] since the former Directive 91/414 EC. NN insecticides are EU crop protection a.s. derived from natural nicotine (Figure 1) used by plants to fight insect pests.

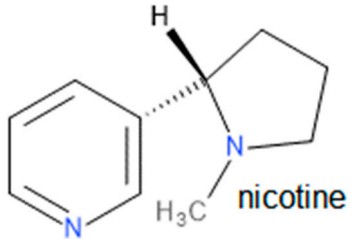

**Figure 1.** Nicotine structure.

These Neonicotinoids insecticides are employed for plant protection for all possible crops against sucking and chewing insects. A specific mode of action (MOA) targeting the nicotinic acetylcholine receptors (nAChRs) is associated with this class of molecules in the synaptic of pest insects with high target specificity for insects, blocking the functional part of the nACh receptor with high affinity [8]. To this respect, flonicamid a pyridinecarboxamide insecticide with a different receptor target (TRPV channels vs nAChRs) was not considered here as an NN even if it is sometimes assimilated to it.

However, some controversial points of view regarding toxicity [9,10] and, moreover, their ecotoxicity [6,7,11,12] regarding this class of pesticides were raised in the late 1990s [13] and later confirmed on bees [14]. Moreover, NN insecticides are already encountering resistance problems [15,16]. Interestingly, nicotine (Figure 1) residues are still resolutely monitored, and its maximum residue limit (MRL), managed by Regulation EC 396/2005, is constantly revised and lowered [17], suggesting periodic use. The initial concept during the introduction of NN insecticides was the reduction in quantities by around a ten times factor compared to previous insecticide generations (i.e., carbamates) still existing and not being completely replaced. Average quantities per hectare in agriculture dropped down from around 3 kg/ha for carbamates to 75–150 g/ha for NN insecticides, but the toxicity against pollinators increased correspondingly [18,19] involving major concerns. If the initial hypothesis of quantity reduction was fulfilled, even going down to below a hundred grams per hectare, the ecotoxicological characteristics, even very correct for the typical species (i.e., ducks, rats), problems quickly arrived via environmental factors accessible both in pollens and nectars and in seed coating dust, as reported extensively [18,19].

The evolution of these active substances in the EU regarding their plant protection status is described in this study. This work was conducted at various levels: their own evolution in number and their entry/exit, the evolution of their regulatory status (prolongation, renewal, Parts (A, B) of the Implementing Regulation EU 540/2011) [20], and their uses in crop protection together with their MRLs [21] and the current survey. The actual potential increase by new NN members through novel PPP applications for available NN molecules is also discussed.

## 2. Materials and Methods

### 2.1. Legal Support

2.1.1. European Pesticides Database

The raw data were retrieved from the European Pesticides Database v3.1. This database lists all the substances approved as well as those where approval is pending and those not approved [21]. The status of the active substances in this database is crucial since no regulations are involved when a.s. are only subjected to "end of approval" abandoned by the applicant, thus without specific regulation.

2.1.2. Directives and Regulations

Regulation (EC) No 1107/2009 [22] is the main and original document dealing with plant protection products (PPP) and substances (pesticides) since 2009. Implementing Regulation (EU) No 540/2011 [20] is the main companion of the PPP regulation as regards the list of approved active substances. Following this, Regulation (EC) No 396/2005 [23] manages the rules on maximum residue levels of pesticides in or on food as well as plant and animal feed.

Subsequently, all the information on one active substance is centralized in the pesticide database, including review reports that contain the GAP usage tables dedicated to field applications. The evolution of NN a.s. was published in the European Union Official Journal. Raw data were extracted from the European Commission pesticide database rev 3.1 website [21] dealing with the Implementing Regulation (EU) No 540/2011 active substance management [20]. Finally, the EFSA journal was used to verify the presence or absence of NN a.s. in crop production and analyse the monitoring operations for the residues of these NN pesticides.

## 2.2. Definitions

### 2.2.1. General Definitions

All definitions were previously defined for agrochemicals [1].

### 2.2.2. Health and Safety Hazards/Risks Phrases

Hazard statements form part of the Globally Harmonized System of Classification and Labelling of Chemicals (GHS). They are intended to form a set of standardized phrases about the hazards of chemicals globally linked to the toxicity of substances and mixtures.

- H301: Toxic if swallowed.
- H302: Harmful if swallowed.
- H317: Skin Sens. 1
- H319: Causes serious eye irritation.
- H331: Toxic if inhaled.
- H332: Harmful if inhaled.
- H335: May cause respiratory irritation.
- H336: May cause drowsiness or dizziness.
- H351: Suspected of causing cancer.
- H360FD: May damage fertility or the unborn child.
- H400: Very toxic to aquatic life.
- H401: Toxic to aquatic life.
- H410: Very toxic to aquatic life with long-lasting effects.
- H411: Toxic to aquatic life with long-lasting effects.

## 3. Results

### 3.1. NN Active Substances

Of the 17 NN insecticide distinct molecules currently known all over the world, less than half (*n* = 7) have been considered and approved in Europe since 2011. From these seven regulated NN insecticides in Europe, only three are still approved with no actual pending NN a.s. and ten from outside Europe, listed separately in Tables 1 and 2. From the maximum NN a.s. approved Europe (*n* = 7), a majority of them (*n* = 4) are not approved anymore, and only three are still approved and listed as plant protection products (PPPs) in Table 1. However, some uses are still covered by emergency (more than 150 since the end of approval of clothianidin in 2019), although some were invalidated [24] at the European level during court Case C-162/21.

**Table 1.** EU NN active substance.

| EU Active Substance | Type | MRL [1] Annex | Tox Ecotox | Nb of Extension/s.a (2023) | Part (Reg. 540) | End of Approval | AIR Program |
|---|---|---|---|---|---|---|---|
| acetamiprid | cyano imidamide | II | Low ADI H302 H400 | 2 | B | 2033 | III |
| clothianidin | nitro guanidine | II | Low ADI H302 H317 H400 H401 H410 | 2 | A | 2019 | III |
| flupyradifurone | butenolide | II, IIIA | H332 H317 H410 | 0 | B | 2025 | VI |
| imidacloprid | nitro guanidine | II, IIIA | H302 H400 H410 | 1 | A | 2020 | VI |

**Table 1.** *Cont.*

| EU Active Substance | Type | MRL [1] Annex | Tox Ecotox | Nb of Extension/s.a (2023) | Part (Reg. 540) | End of Approval | AIR Program |
|---|---|---|---|---|---|---|---|
| sulfoxaflor | sulfo-ximine | II | H302 H401 H411 | 0 | B | 2025 | VI |
| thiacloprid | cyano imidamide | II | Low ADI H302 H317 H318 H332 H336 H351 H360FD H400 H401 H410 | 4 | A | 2020 | III |
| thiamethoxam | nitroguanidine | II | H400 H401 H410 | 2 | A | 2019 | III |

[1] Legend: MRL annex = maximum residue limit number of Annex in Reg. EC 396/2005; tox: risk phrases at EU classification; Nb of extension/s.a = number of end of approval postponing; Part (Reg. 540) = Part from A to E in Reg. EU 540/2011; AIR program = renewal of approval program number from 1(I) to 6(VI).

**Table 2.** Non-EU NN active substance.

| Active Substance | Type | Tox [1] |
|---|---|---|
| cycloxaprid | nitromethylene | £ |
| dicloromezotiaz | mesoionic | £ |
| dinotefuran | nitroguanidine | £ |
| fenmezoditiaz | mesoionic | H400, H410 |
| flupyrimin | pyridylidene | H315, H319, H351, H361 |
| imidaclothiz | nitroguanidine | £ |
| nitenpyram | nitromethylene | H302, H319 |
| nithiazine | nitromethylene | H302, H312, H315, H319 H332, H355 |
| paichongding | nitromethylene | £ |
| triflumezopyrim | mesoionic | £ |

[1] Legend: tox: risk phrases at EU classification. £: no data found.

This first block of molecules in Figure 2, which are quite similar (some in particular, such as thiamethoxam, which is an intermediate in the synthesis of clothianidin), take up motifs from other molecules, called chemical moieties or functional groups. All EU NNs are planar molecules, except sulfoxaflor, which is chiral, sold as a mixture of two racemic diastereomers.

Other NNs, approved all over the world, are exhibited in Table 2. Although those are known by some pesticide databases, only a few are carried by European firms like fenmezoditiaz. Some were approved by USEPA (i.e., nithiazine, nitenpyram, dinotefuran), while some are largely unknown in Europe. None of them are pending in the EU nor were evaluated or submitted by EU PPP regulations to the EFSA (European Food Safety Authority). These molecules are de facto more elaborate and complicated than the first of the NN family, even if the usual functional groups are found.

This second block of a.s. in Figure 3 resumes without trying too much inventiveness the mix of chemical moieties of the previous molecules in Figure 2. Except nithiazine, which is unfortunately not photostable, and was the first synthetic NN involving to use imitation nicotine as a strategy, imidacloprid was the first commercialized NN. Regarding the optical activity of NN a.s., some molecules are planar, while many are chiral (i.e., paichongding and cycloxaprid) in Figure 3, broadly sold as a rough racemic mixture. The fact that known NN moieties were assembled may be the reason why some NNs from Table 2 were not proposed for approval in Europe as PPPs in order to avoid a few lawsuits. Moreover, not having the toxicological and ecotoxicological data of these molecules in hand, it is

not easy to conclude that it is these properties that explain the lack of appetite for the petitioners to deposit them in Europe. These molecules (Table 2, Figure 3) are de facto more elaborated and complicated than the first of the NN family (Table 1, Figure 2), even if the usual functional groups are found, which are exhibited in Figure 4.

**Figure 2.** Structure of the EU NN active substances.

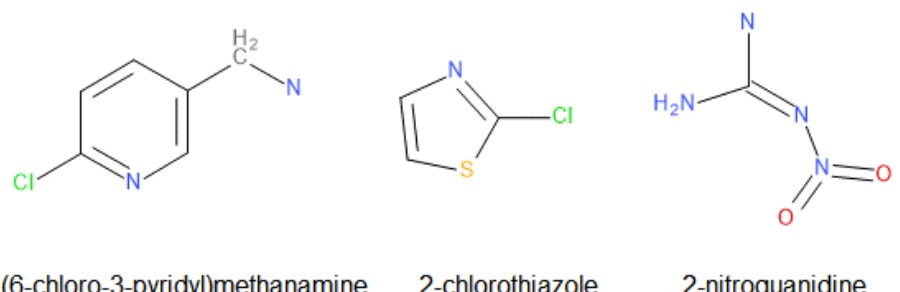

**Figure 3.** Structure of the non-EU NN active substances.

**Figure 4.** Common recognizable moieties or functional groups of the NN active substances.

### 3.2. NN Insecticide Family in Europe

All considered NNs in Europe are listed in Table 1. Surprisingly, none of the 10 other members of the family have actually ever been considered for approval. Some of them are only listed as non-approved in the database [21] with the corresponding remark "Never notified and authorized in the EU". The latter is therefore sought in crop production monitoring plans.

3.2.1. Evolution of the NN Active Substances in Europe since 2011

Of the seven NNs considered in Europe, five were approved in 2011 and two were approved in 2015 (flupyradifurone and sulfoxaflor). Of the five NNs approved in 2011, only one was renewed (acetamiprid), and that is one of the three NNs yet to be approved. None are low-risk or potentially low-risk substances (refereeing to Article 22 of PPP Regulation EU No 1107/2009), although no chemical has been considered thus up to now. However, none of them are currently considered by EU PPP regulation as candidates for substitution (CfSs) (refereeing to Article 24 of PPP Regulation [23]) or on the list of possible CfSs [25]. Their evolution is described in Figures 5 and 6.

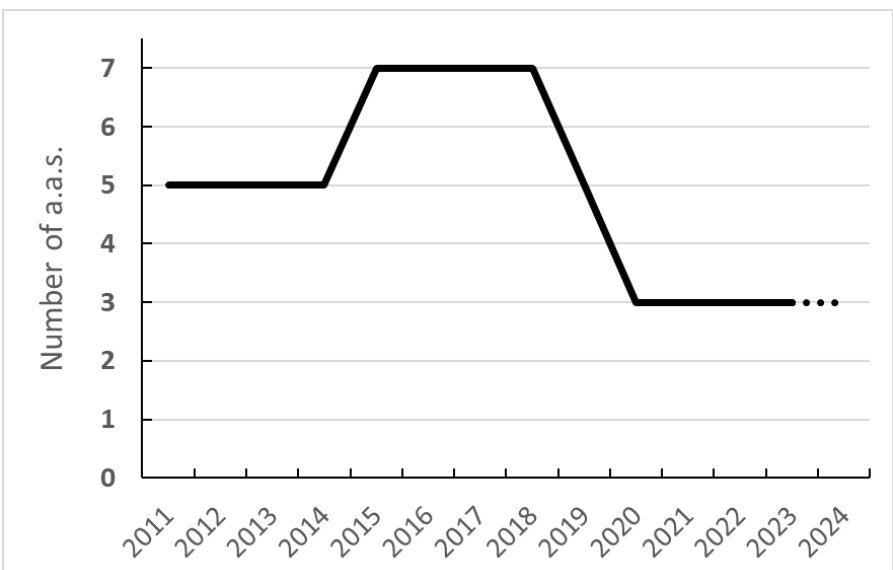

**Figure 5.** Total amount of effective NN active substances in EU.

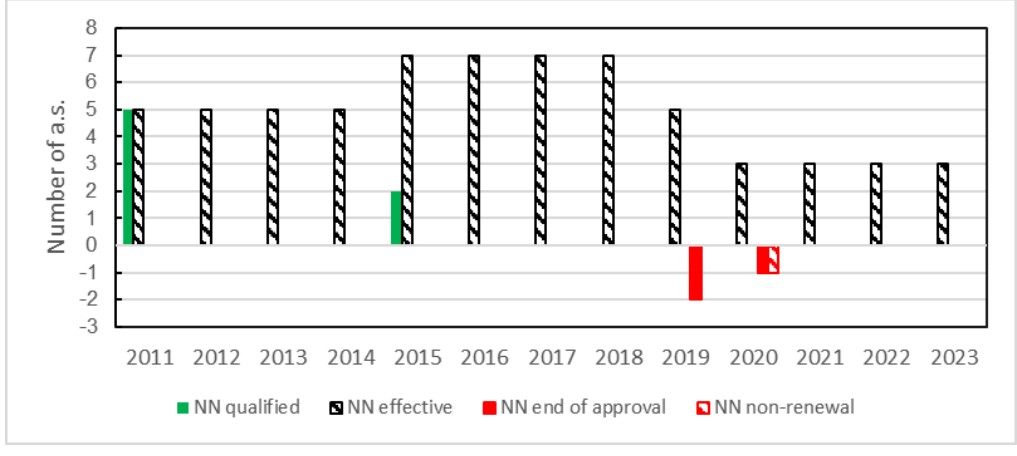

**Figure 6.** Detailed amount of NN active substances in EU.

Figure 5 exhibits only a few variations, and the linear regression shows an unconvincing coefficient. Nevertheless, in percentage terms for NN a.s., the drop is significant (+40% during the first period 2011–2019, and then −57% between 2019 and now). Following, the inputs and outputs of the different NN a.s. with their respective legislative acts are detailed in Figure 6.

Following the modification of substance positions in the different parts of Regulation 540/2011 are exhibited in Figure 7. All remaining NNs are now in Part B since the renewal of acetamiprid in 2018. All further NNs possibly approved will also be listed in this Part B.

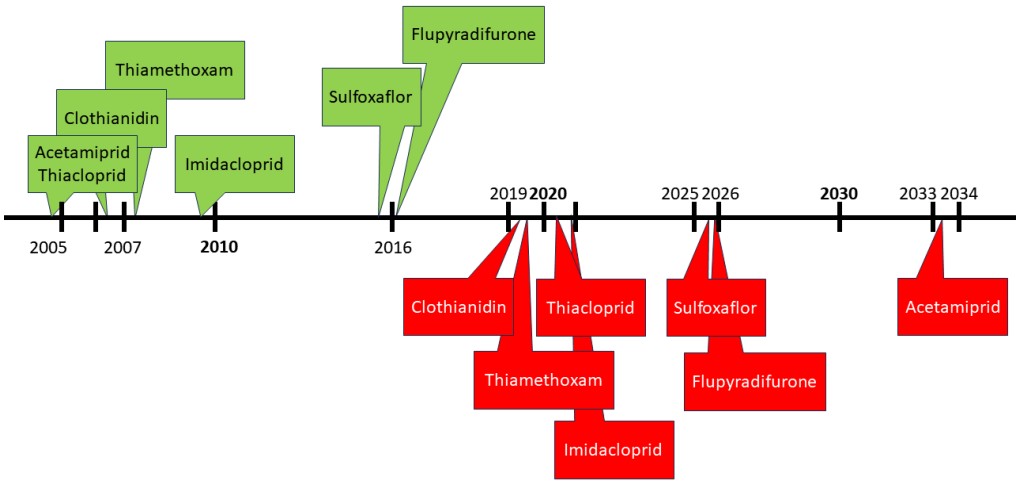

**Figure 7.** Timeframe of regulatory status of NNs in Europe. Approval dates in green; end of approval dates in red.

3.2.2. Consideration of EU NN Crop Usages

All crop classes [2] were covered by NN insecticides, although only a few active substances were dedicated to viticulture (Figure 8). During the last few years, the disappearance of four NNs considerably decreased these usages, down to none for viticulture and cereal.

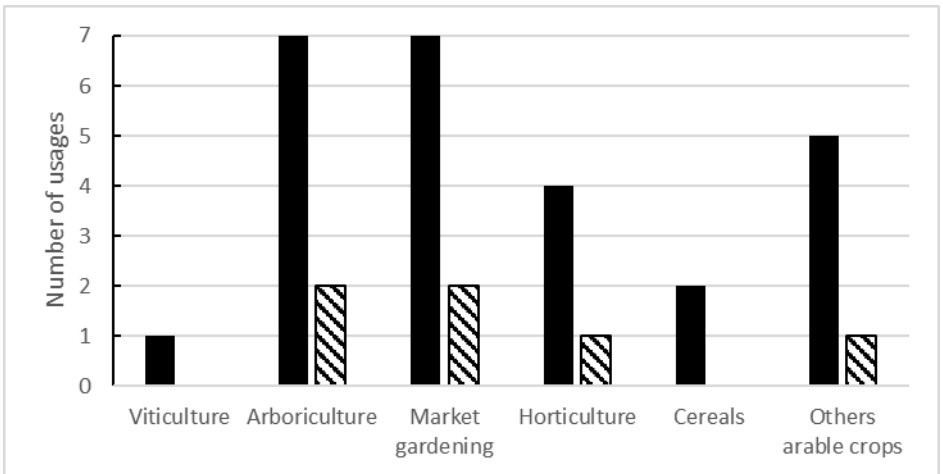

**Figure 8.** Crop usages covered by EU NN active substances. Plain black for all previously approved EU NNs; hatched for actual situation.

We have therefore gone from a maximum of 26 uses (Figure 8 plain black) to a situation showing only 8 crop uses (Figure 8 hatched), a decrease of −70%, of which certain full-field uses have since been banned. Although Figures 2 and 3 are not very explicit, Figure 8 that usage is more convincing for the consequences of the global deletion of not only two

NNs since 2011 but in fact four since the beginning of 2019, not counting the restrictions assigned to the remaining NNs. Following the conclusions of court Case C-162/21 [24], the regulatory situation created will also tend to invalidate any new request for derogation (Art. 53 on emergency of the PPP regulation [23]). This now extends to all non-approved a.s., which currently represents 946 a.s. However, this should not affect derogation requests for new NN candidates for approval.

### 3.2.3. Residues of EU NN

NN insecticide residues are counted for both exposure to humans in food products and non-target species in fields. For food consideration, all NN insecticides have maximum residue limits (MRL) in crop production in Annexes II and/or III of Regulation (EC) No 396/2005 [22], these residues being also of concern for pollinators for some of the NN molecules [26]. For each NN, MRLs are various and different from each crop, available in the pesticide database [21] for each active substance webpage. The current non-approval of some of the NNs will technically lead to eventually switching their MRLs to the default MRL (Annex V of Regulation (EC) No 396/2005) after a few years. These neutral and regulatory data must be correlated with the negative results of investigations at the European level of the presence of residues of NN insecticides in foodstuffs [27], which shows a generalized presence of these a.s. NNs with great quantities, some residues exceeding the ARfD, some outside the authorized EU usage treatments, and only three samples exceeding the MRL (i.e., for acetamiprid and thiacloprid). Regarding these contaminations, the NN a.s. that are still approved (i.e., for sulfoxaflor) are relatively controlled and relieved when necessary [28,29]. For non-target species, residues in crops and soils, which can be mobilized in the next crop by the plants, are also of importance, since they are involved in uncontrolled negative side effects on non-target organisms [30] either for soil organisms, available in soils for plants uptake up to the pollens and nectar, and as seed coating dust liberated during sawing available for contact toxicity [31].

## 4. Discussion

Following this, the evolution of the remaining NNs is subject to regulatory consideration. First, the timing is determined in the programme, and then the applicant has to submit an application; otherwise, the approval is rejected without any other regulatory document at the initial expiry date of the approval execution regulation. As mentioned above, two NN active substances (sulfoxaflor and flupyradifurone) are due to undergo the renewal process over the coming years (Table 1), namely AIR programme VI. Already observable suppressed crop usages are likely to increase in the future considering the actual struggle for chemical PPPs to be renewed [1] while global controversies on neonicotinoids, which started in 1994 [26,32,33], still generate debate and ecotoxicological trials.

The possible evolution of the still-approved NNs in the EU during the re-evaluation procedure is largely dependent on the regulatory life of these active substances and the corresponding PPPs. The first observation from existing NNs is the presence of ecotoxicological concerns for the remaining NNs, which will handicap their renewal. For one of the two NNs approved after the first wave of 2011, sulfoxaflor, recent restrictions [33] for outdoor usage bodes little hope, even if an application has been filed for renewal at the end of 2022 by the petitioner. First of all, indoor usage-restricted businesses may not cover the cost of renewal, or will not be worth it. These restrictions are usual on NN a.s. as already applied on former approved NNs in the EU [34–37] and will remain without a doubt.

The theoretical corresponding situation is exhibited in Figure 9 compared to the previous maximum of seven usages. The possible loss of sulfoxaflor would lead to a reduction in NN to two a.s. in 2026.

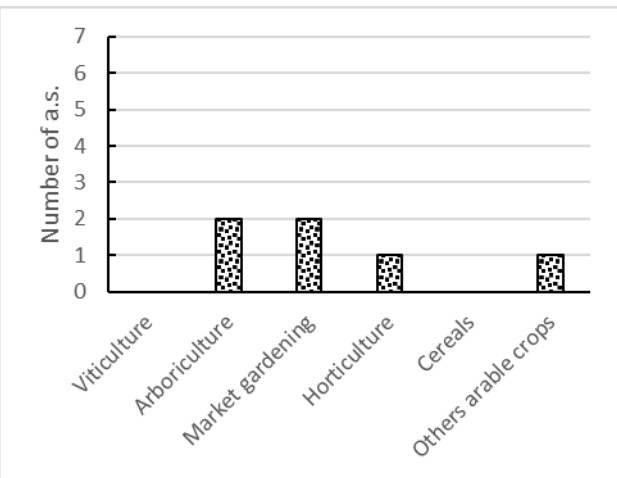

**Figure 9.** Hypothetical crop usages covered by EU NN active substances without sulfoxaflor.

Globally, the total amount of NN a.s. has been decreasing since the transfer of four NN a.s. from former Directive 91/414/EEC despite some new approvals. A global worldwide pool of NN candidates exists but has not yet been exploited, although fenmezoditiaz or triflumezopyrim are supported by occidental firms. Moreover, since no existing NNs are pending nor under evaluation by the EFSA, new approvals in the EU, despite 10 a.s. applications being possible, does not suggest that this family will grow in the future, and even less quickly given the substantial approval duration (in years). In the meantime, it would not be surprising that both NNs (flupyradifurone and sulfoxaflor), still approved and under renewal, will see their expiration date pushed back by a few years as the current process takes time [1]. Moreover, as suggested above, numerous and different combinations of functional groups have already been tested in the twenty or so molecules marketed throughout the world, and undoubtedly many more that are tried are ineffective or possess a toxicologically unfavourable profile. It is undoubtedly becoming complicated to imagine new "similar" structures targeting the nAChR receptor. However, alternatives are already being evaluated [38]. Thus, the possible evolution of the NNs in Europe during ongoing re-evaluation is most likely a reduction to two NNs and possibly less. We must also imagine that these two substances will be studied for their endocrine-disrupting natures. This will undoubtedly lead to an extension of the approval for one or two years with a postponement of the actual end date of these substances until after 2025.

Finally, the legal disappearance of many NN was not only contested by applicants but was also overtaken by numerous Emergency Authorisations (art. 53 of PPP Reg [23]). Almost a hundred of these derogations have been granted since the end of approval of imidacloprid at the end of 2020 (Figure 7)), themselves challenged at the level of the highest European authorities [24]. Therefore, it seems difficult to imagine that new derogations will be granted now on these non-approved NN active substances, deferring these requests to the other approved NNs. This is indeed the case with almost 80 emergency authorisations for the NNs still approved since the beginning of 2021.

## 5. Conclusions

Following the arrival of four NNs during the implementation of the new PPP regulation in 2011, which experienced a rapid increase in 2015, NN active substances are only decreasing in number, uses, and the possibility of use on the ground (constraints, impossibility of derogation and emergency). Moreover, the global reserve of NN active substances has not been exploited, no doubt correlatively, reinforced by the fact that no chemical molecule has been approved since 2019 in Europe [1]. The future of this family of substances with this similar mode of action therefore seems compromised despite great expectations; some applicants placed a lot of hope in it, to the point of naming it the "CNI, chloronicotinyl insecticide" family [39]. These conclusions are supported by the constant

and targeted pressures on this class of observable molecules at the European level [40], as well as the constant tightening of the levels of requirements during approvals and renewals, as well as the implementation of the SUD directive. The two remaining approved NNs under renewal evaluation for 2025, with applications supported and deposited by applicants, will determine the ability of this family to stay active.

**Funding:** This research was funded by national funds of the French Ministries of Ecology and Agriculture through Office Français de la Biodiversité/Ecophyto (XP-BC ACTA 2017-022) and Plan de Relance (ABAPIC ACTA 2021-123, RACAM ACTA 2021-219).

**Institutional Review Board Statement:** Not applicable.

**Informed Consent Statement:** Not applicable.

**Data Availability Statement:** Nonpublic data.

**Conflicts of Interest:** The author declares no conflict of interest.

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
