# Peer review of "Regulatory Evolution of Neonicotinoid Insecticides as Plant Protection Active Substances in Europe"

_agrochemicals, doi:10.3390/agrochemicals2030025_

Round 1

Reviewer 1 Report

1. line 15 and 16 needs improvement in abstract 

2. line 18-20 these correspond to results as well as conclusion should be narrated separately.  

3. introduction is to short not support the study add more references . 

4.Hypothesis is not written clearly Rewrite the hypothesis.

5. Recheck the suitable statistical design. 

6. Figure 6 is not clear differentiate with different color  

7. In figure 7 combine the data to differentiate the differences.  

8. Remove heading from the discussion section. Discussion must be written in paragraph according to your parameter. In discussion section no needs of graph. graph should be aded in results.  

9. conclusion are not match with the results must improved it and rewrite again.

10. References must be according to the journal style 

Moderate editing of English language required

Author Response

  1. line 15 and 16 needs improvement in abstract:
  • Considered, abstract modified and amplified.

  1. line 18-20 these correspond to results as well as conclusion should be narrated separately.
  • Reorganized

  1. introduction is too short not support the study add more references.
  • Considered, introduction modified and amplified

  1. Hypothesis is not written clearly Rewrite the hypothesis.
  • Updated

  1. Recheck the suitable statistical design.
  • Regression suppressed

  1. Figure 6 is not clear differentiate with different color.
  • Figure 6 modified, colored

  1. In figure 7 combine the data to differentiate the differences.
  • Modified

  1. Remove heading from the discussion section. Discussion must be written in paragraph according to your parameter. In discussion section no needs of graph. graph should be aded in results.
  • Modified accordingly

  1. conclusion are not match with the results must improved it and rewrite again.
  • Update of the conclusion.

  1. References must be according to the journal style
  • Complete update of the references.

Reviewer 2 Report

The manuscript entiled :”Regulatory Evolution of neonicotinoid insecticides as plant protection active substances in Europe” by Marchand addresses the current status of neonicotinoids in Europe, from a regulatory perspective. The manuscript would be worthy of interest as the state of the art of one of the most controversial but successful classes of insecticides used worldwide. However, this manuscript should be improved and supplemented in several parts.

-The ecotoxicological perspective is completely neglected; I would suggest adding a section to the results describing the toxicity evidence not only in terms of acute toxicity, but also taking into account potential issues related to chronic exposure, at the population and community level.

-A diagram summarizing the regulatory status and main effects with a timeline of history would greatly improve the quality of the manuscript.

- Persistence/contamination information is not available in the manuscript, which should be incorporated.

Abstract: It would be helpful to include the main concerns about this class of compounds already in the abstract.

Keywords: I woud suggest to replace keywords already present in the title (plant protection products (PPP); neonicotinoid; insecticides) with ecotoxicology, risk assessment, agrochemicals.

Introduction:

L34: remove “of” before class

The introduction needs to be supplemented with the main concerns about the use of neonicotinoids, please expand from L37.

L 39-40 I do not see the need to write the numbers both in the extended world and with Arabic symbols, please make a choice.

The purpose of the study is not entirely clear to me, is this a historical or legal document? I would suggest including more scientific aspects, related to side effects and risks to the environment.

Materials and Methods:

L. 53 not clear to me the meaning of the statement, no regulations are involved because the substances are no longer allowed for use. Reword the period if emergency covered use is intended.

Results:

Table 1: I would suggest including MRLs for each active substance; the document would facilitate the availability of this information by making it clear to readers, instead of just citing external references, such as the Annex in Reg. EC 396/2005.

The section on the structure of the NN active ingredients should be supplemented with references to potential binding nonspecificities or undesirable effects that have led to the need to remodel the structure. This consideration would also help explaining in the discussion why new NN candidates have not yet been evaluated for approval.

Fig. 7 please, add A and B to the figure

Information regarding the use and reduction in the amount of NNs in Europe would benefit from additional information on sales in terms of money invested in purchasing pesticides based on this class of compounds.

Discussion:

The discussion needs to be supplemented with considerations of ecotoxicology and comparative considerations with other compounds (e.g., pyrethroids) that show the efficacy or incidence of resistance selection, clarifying why this class is difficult to replace and having the review emend an examination of both pros and cons.

Author Response

Reviewer #2:

The manuscript entiled :”Regulatory Evolution of neonicotinoid insecticides as plant protection active substances in Europe” by Marchand addresses the current status of neonicotinoids in Europe, from a regulatory perspective. The manuscript would be worthy of interest as the state of the art of one of the most controversial but successful classes of insecticides used worldwide. However, this manuscript should be improved and supplemented in several parts.

-The ecotoxicological perspective is completely neglected; I would suggest adding a section to the results describing the toxicity evidence not only in terms of acute toxicity, but also taking into account potential issues related to chronic exposure, at the population and community level.

  • This is well documented by ref 18 and 19 and references therein, including our own previous work, no need to make plagiarism or over self-citation.
  • Work of the NN Task force is often updated [7],
  • More recent references added

-A diagram summarizing the regulatory status and main effects with a timeline of history would greatly improve the quality of the manuscript.

  • Considered, added

- Persistence/contamination information is not available in the manuscript, which should be incorporated.

  • References added (i.e.33)

Abstract: It would be helpful to include the main concerns about this class of compounds already in the abstract.

  • Considered

Keywords: I would suggest to replace keywords already present in the title (plant protection products (PPP); neonicotinoid; insecticides) with ecotoxicology, risk assessment, agrochemicals.

  • Considered, modified

  1. L34: remove “of” before class
  • Removed

The introduction needs to be supplemented with the main concerns about the use of neonicotinoids, please expand from L37.

  • Considered

  1. L 39-40 I do not see the need to write the numbers both in the extended world and with Arabic symbols, please make a choice.
  • Reorganized, modified

The purpose of the study is not entirely clear to me, is this a historical or legal document?

  • This review is both an historical evolution with the legal status of the NN through the

I would suggest including more scientific aspects, related to side effects and risks to the environment.

  • This is well documented by ref 18 and 19 and references therein, including our own previous work, no need to make plagiarism or over self-citation.

  1. Materials and Methods:
  2. 53 not clear to me the meaning of the statement, no regulations are involved because the substances are no longer allowed for use. Reword the period if emergency covered use is intended.
  • Modified, rewritten

  1. Results:

Table 1: I would suggest including MRLs for each active substance; the document would facilitate the availability of this information by making it clear to readers, instead of just citing external references, such as the Annex in Reg. EC 396/2005. The section on the structure of the NN active ingredients should be supplemented with references to potential binding nonspecificities or undesirable effects that have led to the need to remodel the structure. This consideration would also help explaining in the discussion why new NN candidates have not yet been evaluated for approval.

  • Updated
  • MRL are various and different from each crop, accessible at available in Pesticide database for each active substance
    • e. https://ec.europa.eu/food/plant/pesticides/eu-pesticides-database/start/screen/mrls/details?lg_code=EN&pest_res_id_list=775

Fig. 7 please, add A and B to the figure Information regarding the use and reduction in the amount of NNs in Europe would benefit from additional information on sales in terms of money invested in purchasing pesticides based on this class of compounds.

  • Figures merged
  • Business data are not available

  1. Discussion:

The discussion needs to be supplemented with considerations of ecotoxicology and comparative considerations with other compounds (e.g., pyrethroids) that show the efficacy or incidence of resistance selection, clarifying why this class is difficult to replace and having the review emend an examination of both pros and cons.

  • Discussion amplified but the subject is not that point, although some data may allow comparison between NN. Lot of papers referenced in the manuscript are about ecotox,
  • Comparison with other insecticides or potential substitution substance is not the subject of this manuscript.

Round 2

Reviewer 2 Report

I would like to kindly point out that the changes in the new version are not properly marked. I was forced to compare the two versions to track down the changes. 

In addition, the authors' responses are rushed and unsatisfactory, ruling out potential additions with the statement "not the subject of this manuscript" without argument. This does not seem appropriate to me, especially when the work presented concerns the history and legislation of agrochemicals, which is inevitably linked to the toxicological effects that determine the fate of a chemical in terms of approval.

Often the author has added a bunch of references without expanding on the description in the text; the reader is assumed to read the entire bibliography to get an idea, when the reference should at least be presented in the text. 

However, the main purpose of the manuscript is limited to the historical excursus of NN, so I would consider accepting the work, deeming the discussion partial and not complete in any case.

In addition, I would like to point out that the new Fig. 7 needs to have color legend as in Fig. 6.

Author Response

  • Reorganized,
  • We found out that revision 1 provided was not the right one.
